# Effective Data Utilization in the Context of Industry 4.0 Technology Integration

**Samuel Janík, Peter Szabó \*, Miroslava Mĺkva and Martin Mareček-Kolibiský**

Institute of Industrial Engineering and Management, Faculty of Materials Science and Technology in Trnava, Slovak University of Technology in Bratislava, Jána Bottu č. 2781/25, 917 24 Trnava, Slovakia
* Correspondence: peter.szabo@stuba.sk

**Abstract:** We are part of the digital transformation of society and industry. The digital transformation of industry is based on new technologies brought about by the fourth industrial revolution. The Internet of Things (IoT), Cloud Computing, Cyber–Physical Systems (CPS) and Big Data provide the digital link between machines and individuals in processes. This completely new system is based on efficient data collection, data analysis and immediate interventions in organizational processes based on the results of the analysis. Smart organizations are driven by data and not by models. By working efficiently with the vast amounts of data available, the smart organizations of the future can ensure business sustainability, increase competitiveness through process optimization and reduce costs. In general, the aim of this paper was to identify the means to achieve a paradigm shift from traditional organizations to smart organizations through the use of data in the context of integrating Industry 4.0 technologies. The aim of the research was to determine the extent to which different Industry 4.0 technologies are applied in the effective use of data from specific activities/processes in industrial organizations to bring about a paradigm shift from traditional organizations to smart organizations. The first part of the paper describes the theoretical background of the transition from traditional to smart organizations using selected Industry 4.0 technologies. The second part of the paper characterizes the research objective, the methods used in the paper and the basic statistics used to determine the research questions and hypotheses. The next section evaluates the research questions and hypotheses that were used to meet the research objective. The last part of the paper is a summary of the obtained results, based on which we conclude that the primary challenge for organizations in the Slovak Republic is to learn how to work with the collected data, the need for their appropriate structuring and subsequent archiving, which is manifested by the need for training and application of data analysts in a broader context.

**Keywords:** industry 4.0; big data; IoT; cloud computing; CPS; data driven

## 1. Introduction

Digitalization plays a key role in Industry 4.0 and has enabled a paradigm shift in industrial production through the synergy of the Internet and new technologies. The primary idea behind digitalization is the integration of isolated systems and the easy accessibility of data and information, which is expected to lead to faster data utilization in organizations. One of the ideas behind this Industry 4.0 concept is that by connecting machines, systems, and activities, organizations can create digital-intelligent networks across the value chain that will independently manage processes across the organizations and transform themselves into intelligent organizations [1,2]. In their research, Gürdür et al. [3] stated that digitalization enables the improvement and transformation of activities, functions, models, and processes in organizations and connects the real and virtual world of business through information and communication technologies.

Driven by digitalization, Industry 4.0, and overall advances in cloud computing, IoT, Big Data, CPS and sensors, data are being generated at an unprecedented speed.

Organizations must therefore look for new ways to extract value from these data and use it to give the organization a competitive advantage [4].

The authors of [5] suggested that the industrial organizations of the future will be more than just a system in which production with resources is interconnected and exchanges information. Intelligent industrial organizations will be managed by a fully integrated intelligent system that can manage the entire production system, anticipate problems before they arise and define the method and the time needed to solve them. They can be regarded as organizations that use smart manufacturing in alignment with Industry 4.0 technologies [6]. The Fourth Industrial Revolution is moving towards a complete change in the traditional manufacturing processes of organizations through the integration of Fourth Industrial Revolution technologies into already established manufacturing processes [7]. This revolution has also accelerated the integration of information technologies into manufacturing systems, and the data possessed by organizations are becoming increasingly sophisticated in terms of volume, variety and velocity [8].

Industry 4.0 focuses on intelligent processes within the production system, and these intelligent processes are enhanced by digital technologies such as IoT, Big Data, CPS and Cloud Computing providing continuous data collection and real-time monitoring of individual processes using sensors. This whole system of process monitoring, data collection and processing is mainly executed using the internet and transferring this collected data to large storage within the organization [9,10].

Authors Kufner et al. [11] in their research stated that it is necessary to continuously combine real-time data flows from production to organization systems in order to autonomously respond to the results of predictive analyses. We agree with this idea and are further aiming to develop it in the context of Industry 4.0 as Industry 4.0 brings new solutions that should be flexible and efficient to cope with the increasing volume of data. Kufner also pointed out that these new technologies and solutions should be highly adaptable.

The Industry 4.0 paradigm enables machines to communicate with each other and utilize all the data inside and outside the production processes. This approach allows machines and equipment to operate autonomously and efficiently. It also streamlines communication between machines and people, regardless of their location, and allows multiple devices to work interactively. Through these interactions, all participants in the organization's environment (suppliers, manufacturers and consumers) can obtain and collaboratively analyse data about products and the manufacturing and supply chain processes [12].

In order to fulfil the vision of Industry 4.0 and the transition of current conventional manufacturing processes to intelligent ones, it is necessary that individual systems, subsystems and technologies are fully integrated and, where appropriate, take advantage of the opportunities offered by the digital twin and integrate these systems also within the digital environment. The digital twin model reconstructs physical objects by integrating various data relating to the entire life cycle of the object (Figure 1) [13,14].

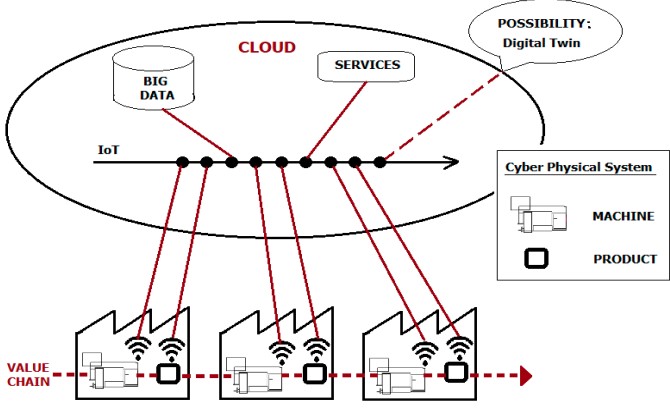

**Figure 1.** Integration of Industry 4.0 technology with possible Digital Twin utilization (Own processing [13]).

The evolving process of the Fourth Industrial Revolution will lead to networked production systems with a high degree of autonomy as well as the ability to self-optimize [15]. The goal of data-driven smart organizations is to use a system that can respond flexibly and adapt quickly to changes by using automation and managing its resources efficiently with minimal human intervention [5].

### 1.1. Data-Driven Smart Manufacturing

Smart manufacturing, or smart production, is based on the effective interconnection of smart technologies with manufacturing technologies. Smart manufacturing embodies new paradigms of thinking within manufacturing technologies that are focused on automating production while taking advantage of the data provided by the information technologies of the Fourth Industrial Revolution. The key technologies of the smart manufacturing concept can include the Internet of Things (IoT), Cyber–Physical Systems (CPS), Cloud Computing and Big Data, among others [16,17].

The utilisation of data has started to play an important role since the beginning of the Fourth Industrial Revolution. Data-driven decisions differentiate modern, intelligent manufacturing from traditional manufacturing in that decisions are made based on facts and data rather than based on theoretical models, opinions, and assumptions [17].

Data from manufacturing processes at the smart manufacturing organization level are obtained by accessing the entire product lifecycle information, from explicit data such as material properties, process temperature and vibration to implicit data, which can include supply chain resources and customer requirements. The premise of a successful product sale is to offer the customer what he both expects and needs. The volume of data that is generated during manufacturing processes in the context of the Fourth Industrial Revolution is growing exponentially [17,18]. Smart manufacturing aims to leverage the data collected throughout the product lifecycle to achieve positive impacts on all aspects of the manufacturing process and the customers within it [16].

At present, researchers are becoming more aware of the value of data, its collection and analysis and are trying to identify ways to use it effectively within manufacturing processes. Figure 2 represents the current process of acquiring data from manufacturing processes and analysing it for the optimal decision toward the optimization of the manufacturing system. This cycle lays the foundation for a data-driven manufacturing paradigm as opposed to conventional model-based manufacturing [17]. Systematic analysis of manufacturing data has the potential to lead to decisions that can improve the efficiency of smart manufacturing [16]. The main difference between the conventional manufacturing process and the modern/smart manufacturing process can be observed in Figure 3. Conventional manufacturing processes in the context of automation can be seen as model-based manufacturing. Experts gain experience through physical observation. Based on this experience, physical models are created through experimental and numerical methods. Such models have added value but their limitation is the extent of efficiency and accuracy as a consequence of the human factor in the process. Intelligent manufacturing that is data-driven makes full use of all manufacturing processes and their real-time data to achieve system flexibility and autonomy [17].

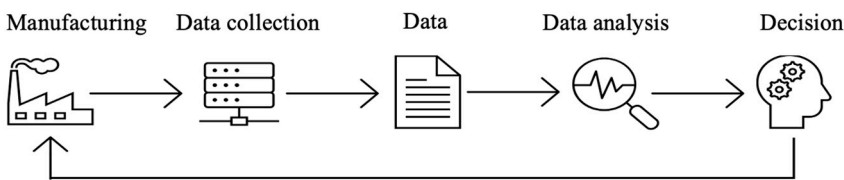

**Figure 2.** Data collection process (Own processing [17]).

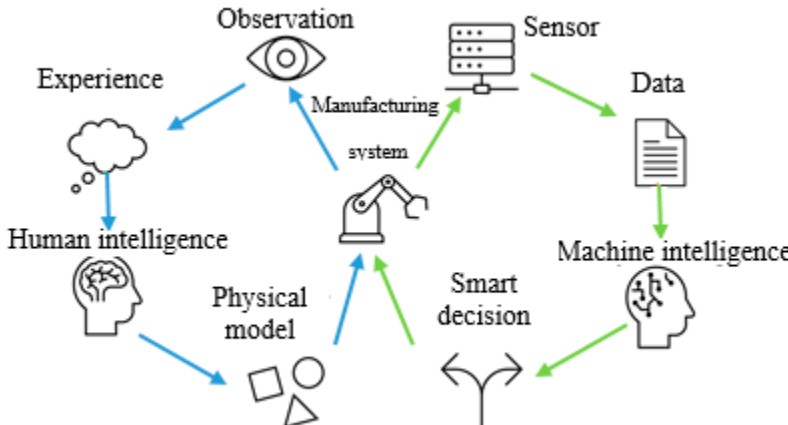

**Figure 3.** Model-controlled production process vs. data-driven production process (Own processing [17]).

Data have long been regarded as an important part of manufacturing and manufacturing processes that are data-driven [19]. Therefore, it can be concluded that data-driven manufacturing is a necessary prerequisite for smart manufacturing. Data are becoming a key factor for increasing competitiveness and organizations are becoming aware of its great strategic importance in the long term [16].

With the emergence of the Fourth Industrial Revolution, manufacturing systems are being transformed into digital ecosystems. Big Data and Internet of Things (IoT) technologies play an essential role in this transformation. With the advent of this revolution, industrial organizations have entered a new age of 'Big Data', where the volume, velocity and variety of data are growing exponentially. The Internet of Things is based on interconnected objects, machines, technologies and people, which directly support the concept of Big Data by enabling the collection of even more data and information [20].

*1.2. Internet of Things (IoT)*

The Fourth Industrial Revolution is pushing towards intelligent organizations, where communication and access to the information within an organization will not only take place between people. Machines, on behalf of humans, will seek to communicate with other machines and collect the necessary data. All this communication will take place via the Internet, called the Internet of Things, whose architecture is technically based on data communication tools. All objects (human or machine) defined in the Cyber–Physical system will use the Internet of Things to communicate [5].

The Internet of Things (IoT) can be characterized as the outcome of the radical evolution of the Internet connected to a network of interconnected objects that create an intelligent environment. The IoT is a system of interconnected computing devices that are equipped with unique identifiers and the ability to transport data within a network without the need for human-to-human or human-to-computer interaction, according to the technical definition. The devices can either be mechanical or digital. The basic idea of IoT is to take all physical devices around the world and connect them with the internet [21].

IoT enables the connection of people and objects anytime, anywhere, with anything and anyone by the exploitation of any path or network. The designation for IoT application in industrial organizations is called IIoT and refers to integrating IoT into the industry by connecting sensors, control systems, machines, devices, Big Data analysis and security. By utilizing IIoT, industrial organizations are able to become intelligent based on optimizing processes and resources and creating autonomous systems [20].

IoT can be described as a network comprising several connected nodes that depend on sensory, transmission and information-processing technology that communicate with each other as smart components to achieve a specific goal using the Internet as a communication medium with no time and space constraints [22].

The Internet of Things extends the application of technologies such as RFID to tools, technologies and components in manufacturing processes, enabling communication and information sharing between them. RFID is a wireless technology that enables the automatic identification of objects at a distance [23]. Every physical object can become intelligent thanks to IoT, and the data obtained from intelligent objects can assist organizations in optimizing processes at all levels of the organization [24].

IoT strategically promotes the use and effective integration of sensors into manufacturing processes to collect real-time data, thereby contributing to data-driven manufacturing [16].

### 1.3. Cloud Computing

Cloud computing, in collaboration with IoT, enables the storage, processing and analysis of vast amounts of data, thereby contributing to the realization of large-scale simulations of various aspects of industrial organizations [6,16]. It is a flexible and inexpensive technology that provides services including infrastructure, hardware, software, platforms and other information within information technology. Users can use the provided services by utilizing the application through computers and storage servers [5]. Cloud technologies serve as tools to enable worldwide access to information [20]. Legashev [25] in his research stated that cloud computing is one of the most used technical solutions for data collection, data processing and its subsequent use in the business environment.

Several major cloud computing platforms can be used in industrial IoT and Big Data analysis. The key features of these platforms are security, data storage capacity and flexibility for client requirements [26]. By using the services of cloud platform providers, industrial organizations are not forced to acquire their own IT infrastructures for data storage. By shifting the responsibility to external companies, industrial organizations are able to reduce their operational costs, increase security and gain the flexibility to scale up IT infrastructures based on business needs [27].

Cloud Computing is a technology that provides reliable storage and management of data thus providing opportunities to exploit the potential offered by the Internet of Things and Cyber–Physical Systems [24].

### 1.4. Cyber–Physical System (CPS)

Cyber involves computation, communication and control that is discrete, switched and logical. Physical represents natural, man-made systems that are governed by the laws of physics and operate continuously. A Cyber–Physical system (CPS) connects information technologies to the physical (mechanical, electronic) world (objects). The communication between them takes place by using data infrastructure such as the Internet [6].

According to [6], CPS consists of two main functional components:

- Advanced connectivity, which provides real-time data acquisition from the physical world while providing information feedback from cyberspace;
- Intelligent data management, computational and analytical capabilities that create cyberspace.

Technologies that are closely connected to CPS include IoT, Cloud Computing and wireless sensors, which are considered an important part of CPS [6,13] while in their integrity they can be defined as the centre of Industry 4.0 innovation [28]. More devices, machines, vehicles and production tools are now embedded with sensors [20].

Cyber–Physical systems connect the physical world with the virtual world of information technology and software through the use of data, communication, the Internet and services. The application of CPS by leveraging the Internet of Things in industrial organizations contributes to the generation of a large amount of information that is necessary yet complex to analyse. For the effective analysis of a large amount of data, Big Data analysis is used through which meaningful outputs can be obtained and the real value of Big Data can be uncovered [20].

## *1.5. Big Data*

Through the rapid development of the Internet and Industry 4.0 technologies, vast amounts of data and information are being generated and collected. Their processing and analysis go beyond the capabilities of traditional tools. Big Data allows the quick and efficient management and use a of constantly growing database. Big Data technology enables the analysis and separation of important information, and data from less important ones—it helps to draw conclusions and supports efficient knowledge transfer to achieve strategic goals. With this technology, data that have been collected in various incompatible systems, databases and websites are also processed and combined to provide a clear picture of the situation [29]. Big Data enables businesses to optimize processes, improve customer service, and offer a starting point for developing new business models [30]. Big Data refers to a set that collects a large amount of data. This amount of data is difficult to process using traditional storage, applications, and processing methods. By introducing Big Data technology, it is possible to perform tasks that involve a huge amount of data and manage, control, and implement improvements more efficiently and in real-time. Big Data technology is able to analyse a huge amount of different data at an advanced level that flows in the system at a high speed, and thus Big Data is a pillar of the Industry 4.0 concept because the functionality of the Fourth Industrial Revolution concepts that are based on data are derived from the rapid analysis of data [31].

A Big Data solution package can consist of several critical components, from the hardware layer (such as storage, computing and networking—IoT and cloud computing) to the analysis layer (the actual processing of the data—Big Data analysis), where business statistics are generated using improved statistical and computational methods, algorithms and data visualization [32]. In addition to big data, many other features define the difference between Big Data and terms such as "massive data" or "very big data". These differences and different concerns in science and technology aspects cause different definitions of Big Data.

Big Data can be simplistically described as a set of data that could not be collected, managed, analysed and processed by conventional computers within an acceptable scale and capacity. This means that the volumes of data sets conforming to the Big Data standard change and increase over time or with technological improvements. According to [33], in line with this definition, Big Data represents another possible development for innovation, competitiveness and productivity of industrial organizations.

At the core of the Big Data paradigm is the extraction of knowledge from data as the basis for intelligent services and decision-making systems. This idea covers many scientific disciplines and explores various techniques and theories from different fields including data mining, machine learning, information discovery, analysis generation, etc. Big Data is undoubtedly becoming an important trend in process optimization. The results of optimization depend on the accuracy and availability of information. The concept of Big Data provides analytics to enable insights into the customer's experience in the process, their requirements, and the quality of the products [33,34].

### 1.5.1. Data Collection and Data Mining

Modern manufacturing systems are equipped with advanced sensors that collect sequential data from different processes. These data are low in volume when accessed individually, but as a whole, they generate a huge amount of data. To generate simulations, evaluations and predictions, it is fundamental that high-quality data collection becomes a priority goal for organizations in their transformation into smart organizations [17].

Data mining is the process of analysing data to identify new patterns within one or more data sets. Data mining techniques use integrated data from a database by utilizing statistical and mathematical methods [35]. Some advanced data mining methods use clustering, classification, association rules, regression, prediction and variance analysis. By utilizing such methods, large amounts of dynamic and raw data can be organized and evaluated [16].

### 1.5.2. Big Data Analysis

Big Data offers a huge opportunity to transform the current manufacturing paradigm towards smart manufacturing. Big Data enables organizations to adopt data-driven strategies and to become more competitive [16].

With the development of the Internet of Things, smart manufacturing is enabling the generation of vast amounts of data. Big Data was measured in terabytes in 2005, petabytes in 2010, exabytes in 2015 and zettabytes in 2020. This amount of data brings with it a range of challenges [8,20,27].

The demands for efficiently extracting valuable information from such a large amount of data are constantly increasing. Data analysis techniques can be divided into the following categories:

- System infrastructure;
- Analytical methods.

System infrastructure focuses on preparing data for analysis, while analytical methods focus on how to extract useful information from the data [27].

Continuous production processes, a multitude of sensors and real-time data and their efficient transmission determine the data's basic characteristic called as 3V, which are represented by [5,8,20,27]:

- Volume;
- Variety;
- Velocity.

Big Data analysis can contribute to streamlining production planning and optimizing manufacturing processes. Information obtained from customer orders, the status of production equipment, production capacity, supply chain data, sales and available inventory can be analysed using Big Data analysis methods. Production plans can be created based on the outputs of these analyses, which also ensures the optimal configuration of production resources and processes for the execution of individual tasks. Big Data analysis contributes to the assessment of the optimality of technological processes, by analysing process data, including historical data, and individual relationships that arise within processes and technology parameters, thus ensuring increased productivity, quality and cost reduction [16].

Big Data analysis is becoming an essential requirement for extracting important insights from data, which will enable industrial organizations to reduce their costs and increase their profits. Big Data analysis is based on the principles of extracting, validating, translating and exploiting data. It is an emerging field that focuses on making empirical predictions. Organizations that are data-driven use analytics in decision-making processes at all levels of the organization [19].

Sadiku et al. [19] concluded that industrial organizations are beginning to utilize Big Data analysis because of the following reasons:

- Gaining control over the vast amount of data generated;
- Utilizing these data to support decision-making processes and increase productivity.

Thanks to the use of digitization, Industry 4.0 and overall advancements in cloud solutions, IoT, Big Data, CPS and sensors, data are being generated at an unprecedented rate. Organizations must therefore look for ways to select from these data those that will be of value and through which the organization will gain a competitive advantage [4].

## 2. Materials and Methods

The aim of the research was to determine the extent to which individual technologies of Industry 4.0 are applied in the effective utilization of data from specific activities/processes in industrial organizations to shift the paradigm from traditional organizations to smart organizations. Literary titles used in the theoretical part represent a summary of knowledge from the latest publications registered in reputable databases or published in journals with a high impact factor. The theoretical knowledge was summarized to correspond with the

content of the research part of the paper. As has been described in various literature sources, the application of the different technologies of Industry 4.0 may vary depending on the size of the organization or the focus of the business. The research problem was elaborated through the definition of research questions and research hypotheses.

Empirical data on the research subject were collected using a scientific questionnaire. The questionnaire was developed on the basis of a pilot survey (number of respondents 80—1 expert per 1 enterprise, members of the industrial council and top managers of the participating industrial enterprises) and on the basis of structured interviews with managers of industrial enterprises. The questionnaire contained a total of 37 closed-ended questions, the first part was aimed at finding out the identification and demographic characteristics of the respondents. Due to the narrower focus of the research in question, only a part of the obtained data closely related to the topics of Big Data and CPS was used.

In the elaboration of all parts of the paper, from the Introduction, through the Theoretical Background to the Discussion, basic thought processes such as analysis, synthesis, abstraction, concretization, deduction, analogy, comparison, etc., were used. Data interpretation was approached through descriptive and quantitative statistical methods. The research questions were formulated for the purposes of the above research (see acknowledgements), and the outputs were statistically evaluated separately for publication purposes. Parametric and non-parametric statistical tests were used to test the relationships between dependent and independent variables. The collected data were processed using Microsoft Excel 365 (Academic Licence), IBM SPSS Statistics version 22.0 (Statistical Package for the Social Sciences). The anonymity of the respondents was maintained in the processing of the results.

The research sample consisted of employees of industrial organizations of all sizes in three regions of Slovakia. The sample consisted of 556 respondents. Of this number, 80 (14%) respondents worked in the public and state administration sector, 138 (25%) in the service sector and 338 (61%) in industrial production. Given the research focus of our paper, we examined organisations operating in the industrial sector. A deeper breakdown of the respondents operating in each of the industrial manufacturing sectors is provided in Table 1.

**Table 1.** Individual sectors of industrial production.

| Industrial Production Sectors | Absolute Frequency | Relative Frequency |
|---|---|---|
| automobile industry | 134 | 40% |
| engineering, metalworking and metallurgy and other production | 87 | 26% |
| construction industry | 17 | 5% |
| design and engineering | 11 | 3% |
| electrical engineering and energy | 32 | 9% |
| chemistry and plastics | 25 | 7% |
| transport and logistics | 8 | 2% |
| information technology, telecommunications | 5 | 1% |
| food industry, agriculture and forestry | 15 | 4% |
| other | 6 | 1% |

In addition, from the results of the questionnaire survey, we found that 233 (69%) of the respondents belonged to the category of large organizations (250 or more employees), 64 (19%) belonged to the category of medium organizations (up to 249 employees) and the remaining 41 (12%) belonged to the category of small or micro-organizations (less than 49 employees).

## 3. Research Results

In order to accomplish the aim stated above, the research questions and hypotheses were defined, statistically tested and interpreted.

RQ1: How do organisations perceive the importance of Industry 4.0?

We used basic descriptive statistics to evaluate the research question. In the first research question, we investigated the perception of the importance of Industry 4.0 in organizations. The results obtained are shown in Table 2. Based on these results, it can be concluded that 177 respondents (52%) perceived Industry 4.0 as very important for the future of the organization. The impact of the COVID-19 pandemic increased the perceived importance of Industry 4.0 for another 23 (7%) respondents. As many as 83 (25%) respondents had no awareness of the strategic goals of the organization in terms of the innovation that Industry 4.0 brings, according to the survey results. A total of 55 (16%) of respondents indicated their perception of the importance of Industry 4.0 as currently a low priority. Differences in perceptions of the importance of Industry 4.0 were also significant depending on the size of the organization. Industry 4.0 was particularly important in medium and large organizations. In micro and small organizations, the perception of Industry 4.0 had a low priority.

**Table 2.** The importance of the perception of Industry 4.0 in the organization.

| | Number of Employees | | | | | | | | | |
|---|---|---|---|---|---|---|---|---|---|---|
| | 1 to 9 | | 10 to 49 | | 50 to 249 | | 250 and More | | Overall | |
| | Absolute Frequency | Relative Frequency | Absolute Frequency | Relative Frequency | Absolute Frequency | Relative Frequency | Absolute Frequency | Relative Frequency | Absolute Frequency | Relative Frequency |
| very important for the future of the company | 3 | 19% | 6 | 24% | 30 | 47% | 138 | 59% | 177 | 52% |
| increasing importance due to the COVID-19 pandemic | 1 | 6% | 1 | 4% | 5 | 8% | 16 | 7% | 23 | 7% |
| currently low priority | 6 | 38% | 10 | 40% | 10 | 16% | 29 | 12% | 55 | 16% |
| do not know | 6 | 38% | 8 | 32% | 19 | 30% | 50 | 21% | 83 | 25% |
| overall | 16 | 100% | 25 | 100% | 64 | 100% | 233 | 100% | 338 | 100% |

RQ2: At what stage is the implementation of Industry 4.0 in their organisation?

In the above research question, we investigated the level of implementation of Industry 4.0 in industrial organizations. The results obtained are shown in Table 3. Based on the above results, it can be concluded that 139 respondents (51%) had implemented the technologies of Industry 4.0 for a long time and planned to continue doing so. A total of 60 respondents (15%) expressed that they did not currently implement Industry 4.0 technologies but planned to implement them in the near future. A total of 120 respondents (32%) could not answer the question, which may have been related to the position in which the respondents worked, and whether they had enough information about the issue. Table 3 also includes the responses to each option depending on the size of the business. Industry 4.0 technologies had been implemented for a long time, especially in medium and large organizations.

Based on research questions RQ1 and RQ2, research hypotheses were defined.

**Hypothesis 1.** *There is a correlation between perceived importance and the implementation of Industry 4.0.*

The hypothesis, the correlation between perceived importance and the implementation of Industry 4.0 was verified by the Chi-square test (Table 4) and the strength of the correlation was determined using Cramer's value. The significance came out less than 0.05; that is, we rejected H0 at a 0.05 level of significance and this implies that there was a correlation between perceived importance and Industry 4.0 implementation. According to Cramer's V, the value was 0.392 which means that the dependence between the variables can be considered strong. We also verified the result using the Pearson correlation test which showed that the dependency exists. The said variable was correlated at sig. = 0.05

level with Pearson correlation coefficient value r = 0.517. The significance value reached the required level (sig. < 0.05); therefore, we can confirm that there was a strong correlation between the variables tested.

**Table 3.** Implementation of Industry 4.0 in the organization.

| | Number of Employees | | | | | | | | | |
| | 1 to 9 | | 10 to 49 | | 50 to 249 | | 250 and More | | Overall | |
| | Absolute Frequency | Relative Frequency | Absolute Frequency | Relative Frequency | Absolute Frequency | Relative Frequency | Absolute Frequency | Relative Frequency | Absolute Frequency | Relative Frequency |
|---|---|---|---|---|---|---|---|---|---|---|
| we have been implementing Industry 4.0 technologies for a long time and plan to continue doing so | 3 | 19% | 2 | 8% | 16 | 25% | 118 | 51% | 139 | 41% |
| we are not currently implementing Industry 4.0 technologies, but we plan to implement them | 3 | 19% | 7 | 28% | 14 | 22% | 36 | 15% | 60 | 15% |
| we do not currently implement Industry 4.0 technologies and do not plan to implement them | 4 | 25% | 4 | 16% | 7 | 11% | 3 | 1% | 18 | 1% |
| do not know | 6 | 38% | 12 | 48% | 27 | 42% | 75 | 32% | 120 | 32% |
| other | 0 | 0% | 0 | 0% | 0 | 0% | 1 | 0% | 1 | 0% |
| overall | 16 | 100% | 25 | 100% | 64 | 100% | 233 | 100% | 338 | 100% |

**Table 4.** Results of hypothesis 1 using the Chi-square test.

| | Value | df | Asymptotic Significance (2-Sided) |
|---|---|---|---|
| Pearson Chi-Square | 61.333 [a] | 4 | 0.000 |
| Likelihood Ratio | 55.615 | 4 | 0.000 |
| Linear-by-Linear Association | 23.409 | 1 | 0.000 |
| N of Valid Cases | 200 | | |

[a] Two cells (22.2%) have expected count less than 5. The minimum expected count is 1.43.

**Hypothesis 2.** *There is a correlation between Industry 4.0 implementation with the size of the enterprise.*

The hypothesis, the correlation between Industry 4.0 implementation and organization size, was verified by the Chi-square test (Table 5) and the strength of the correlation was determined using Cramer's value. The significance came out as less than 0.05 which means we rejected H0 at a 0.05 level of significance and this implies that there was a correlation between Industry 4.0 implementation and organization size. According to Cramer's V, the value was 0.312 which means that the correlation between the variables can be considered moderate. We also verified the result using the Pearson correlation test, which showed that there was a correlation between the variables under study. The said variable was correlated at sig. = 0.05 level with the value of Pearson correlation coefficient r = −0.447. The significance value reached the required level (sig. < 0.05); therefore, we can confirm that there was an inverse moderate correlation between the variables tested.

**Table 5.** Results of hypothesis 2 using the Chi-square test.

| | Value | df | Asymptotic Significance (2-Sided) |
|---|---|---|---|
| Pearson Chi-Square | 40.164 [a] | 4 | 0.000 |
| Likelihood Ratio | 36.326 | 4 | 0.000 |
| Linear-by-Linear Association | 38.642 | 1 | 0.000 |
| N of Valid Cases | 206 | | |

[a] Three cells (33.3%) have expected count less than 5. The minimum expected count is 0.88.

Figure 4 represents the percentage of Industry 4.0 implementation depending on the size of the organization. Based on the Chi-square test, the calculated correlation coefficient, and the graphical representation, we can confirm that the implementation of Industry 4.0 depended on the size of the organization. The larger the organization, the more likely it was to implement the different technologies of Industry 4.0.

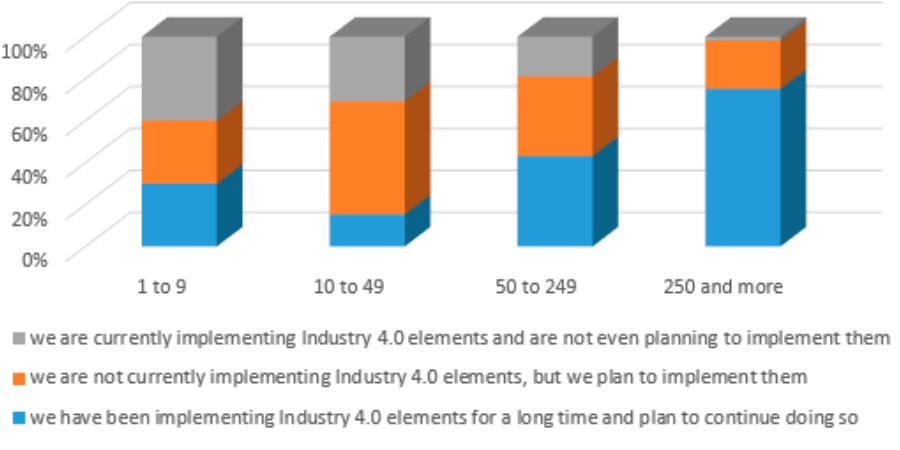

**Figure 4.** Implementation of Industry 4.0 depending on the size of the company.

A uniform deployment of all technologies of Industry 4.0 in the conditions of industrial organizations is neither possible nor reasonable. Organizations usually deploy those parts of Industry 4.0 technologies without which the functioning of smart organization systems would not be possible at any given moment. For this reason, we defined another research question, where we researched which technologies of Industry 4.0 are most frequently implemented in industrial organizations. In this case, it can be assumed that the COVID-19 pandemic has acted as a catalyst in the introduction of technologies related to digitalization.

RQ3: Which technologies of Industry 4.0 are implemented in the organizations?

Based on the responses of respondents from each industrial organization (Figure 5), we identified that the most commonly deployed element was Cloud Computing, indicated by 138 (41%) respondents, followed by Autonomous Robots, indicated by 127 (38%) respondents and Cybersecurity, with 107 (32%) respondents. A surprising result was that the Industry 4.0 Internet of Things (IoT) element was marked by only 45 (13%) respondents. IoT is used to connect Cyber–Physical systems with cloud solutions and we did not assume that organizations did not have their systems interconnected. We can only state that this is an anomaly when respondents did not have enough knowledge regarding IoT and thus could not properly define the operation of this technology within an organization.

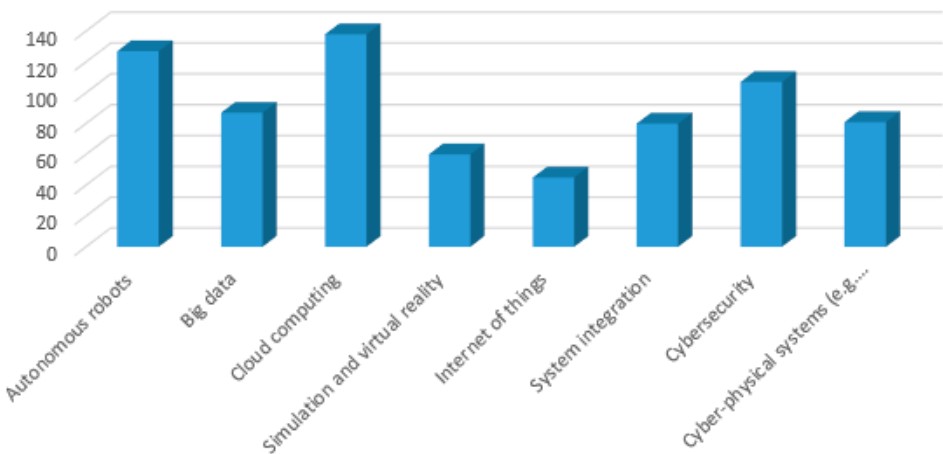

**Figure 5.** Introduced technologies of Industry 4.0.

With regard to hypothesis 2, where we confirmed the correlation of Industry 4.0 implementation with the size of the organization, in Table 6 we have broken down the different technologies introduced depending on the size of the organization. From above Table 6, it is clearly evident that the individual technologies of Industry 4.0 were most often implemented in organizations with 250 or more employees. The introduction of the technologies in the micro-organization was unlikely, as can be seen in Table 6.

**Table 6.** Implementation of Industry 4.0 technologies depending on the size of the company.

| Technologies of Industry 4.0 | Number of Employees | | | | | | | | | |
| --- | --- | --- | --- | --- | --- | --- | --- | --- | --- | --- |
| | 1 to 9 | | 10 to 49 | | 50 to 249 | | 250 and More | | Overall | |
| | Absolute Frequency | Relative Frequency | Absolute Frequency | Relative Frequency | Absolute Frequency | Relative Frequency | Absolute Frequency | Relative Frequency | Absolute Frequency | Relative Frequency |
| Autonomous robots | 1 | 0% | 0 | 0% | 21 | 6% | 105 | 31% | 127 | 38% |
| Big data | 3 | 1% | 1 | 0% | 11 | 3% | 72 | 21% | 87 | 26% |
| Cloud computing | 6 | 2% | 6 | 2% | 37 | 11% | 89 | 26% | 138 | 41% |
| Simulation and virtual reality | 0 | 0% | 4 | 1% | 8 | 2% | 48 | 14% | 60 | 18% |
| Internet of things | 1 | 0% | 2 | 1% | 7 | 2% | 35 | 10% | 45 | 13% |
| System integrity | 1 | 0% | 3 | 1% | 17 | 5% | 59 | 17% | 80 | 24% |
| Cybersecurity | 3 | 1% | 0 | 0% | 25 | 7% | 79 | 23% | 107 | 32% |
| Cyber–Physical systems (e.g., automatic conveyors) | 0 | 0% | 0 | 0% | 15 | 4% | 66 | 20% | 81 | 24% |
| Do not know | 9 | 3% | 14 | 4% | 14 | 4% | 59 | 17% | 96 | 28% |

Digitalization is becoming a crucial factor in increasing competitiveness now and in the future. It is driven by data and leveraging it to its best advantage. By making efficient use of the available data, organizations are able to make decisions and analyses in order to meet their objectives, e.g., reducing costs, improving the use of production capacity, increasing quality, improving product features, etc. Therefore, in the next section, we focus on the current ways/opportunities of data use in organizations (RQ4—see on Figure 6—should correctly be Figure 5).

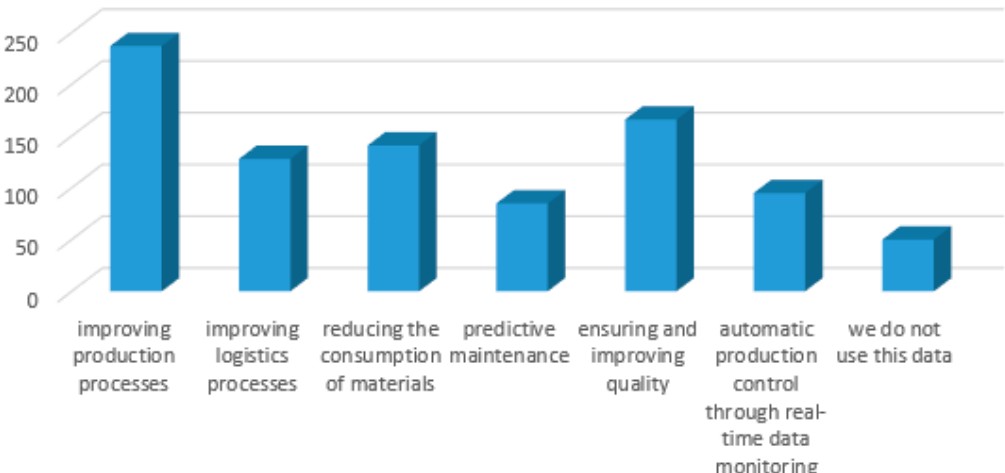

**Figure 6.** Areas of use of data obtained from processes.

RQ4: What the data collected from each process is used for?

The most common response was improving production processes, which was indicated by 237 (70%) of respondents from industrial organizations. The second area in order was quality control and improvement, which was indicated by 166 (49%) of the respondents.

RQ5: How does the introduction of Industry 4.0 technologies affect the data utilization from the analysed processes of the organisation.

Based on the responses from the respondents, Table 7 summarises the different areas of data utilization according to the Industry 4.0 technologies introduced.

**Table 7.** Impact of collected data from selected Industry 4.0 attributes on processes.

| | Improving Production Processes | | Improving Logistics Processes | | Reducing the Consumption of Materials | | Predictive Maintenance | | Ensuring and Improving Quality | |
|---|---|---|---|---|---|---|---|---|---|---|
| | Absolute Frequency | Relative Frequency | Absolute Frequency | Relative Frequency | Absolute Frequency | Relative Frequency | Absolute Frequency | Relative Frequency | Absolute Frequency | Relative Frequency |
| Big data | 71 | 21% | 48 | 14% | 51 | 15% | 36 | 11% | 57 | 17% |
| Cloud computing | 112 | 33% | 66 | 20% | 71 | 21% | 46 | 14% | 85 | 25% |
| Internet of things | 37 | 11% | 24 | 7% | 22 | 6% | 12 | 4% | 29 | 8% |
| Cybersecurity | 87 | 25% | 55 | 16% | 56 | 17% | 46 | 14% | 71 | 21% |
| Cyber–Physical systems (e.g., automatic conveyors) | 74 | 22% | 47 | 14% | 48 | 14% | 34 | 11% | 55 | 16% |

As can be seen from Table 7, the introduction of Cloud Computing and Big Data Processing–Big Data were identified by the highest number of respondents in relation to data utilization within each of the defined areas. Statistical evaluation of the impact of the collected data depending on the degree of implementation of the selected Industry 4.0 technologies was not the subject of our investigation; however, the results presented in Table 7 clearly indicate that the technologies and areas we selected were interrelated and their separate implementation will not bring the desired (or any) effect to the organization.

## 4. Discussion and Conclusions

Data are an important internal resource for any industrial organization. Proper use of high-quality data can assist organizations in analysis, decision making and forecasting to meet strategic objectives. However, the prerequisite is that is the data are properly collected, adequately evaluated and processed into the resulting information that will benefit and add value to smart organizations. Their utilization is not limited, they can help in the field of decision making, optimization, or process improvement, as well as in the application of various lean methods and industrial engineering practices in practice.

Big Data is perceived as a technology for a rigorous and intelligent system for evaluating the data collected. In this system, it is a new more sophisticated data source that can be used to better identify the state inside the processes and understand the value of data that can be obtained through Big Data technologies. The use of Big Data in this context will have a significant impact on creating a competitive advantage for businesses [4]. The importance of Big Data, its collection and use for effective process management was confirmed by the data in Tables 6 and 7 (maintained by the authors of the publication), on the basis of which it is possible to fully agree with Santos et al. The Internet of Things will provide industrial organizations with interconnected manufacturing Cyber–Physical systems, sensors will collect data from processes across the entire product lifecycle, which will be stored on Cloud Computing platforms and evaluated through Big Data analysis to add value to the organizations. This process supports the transformation of traditional organizations into smart ones.

Several barriers can hinder the full implementation of Industry 4.0, such as high costs, financial limitations, lack of management support, resistance to change, poor quality infrastructure, poor quality implementation data and many other negative influences on the concept. Thus, this paradigm relies on digitizing manufacturing processes and entire systems to manage production in real-time with minimal cost. This implies that digitalization will also have a significant impact on manufacturing systems and their sub-systems to be able to respond quickly to customer demands [33,36].

The aim of the research was to determine the extent to which individual elements of Industry 4.0 are applied in the effective use of data from individual processes in industrial organizations. The results of the research confirmed several important facts as follows: there was a dependency between the perceived importance and the implementation of Industry 4.0, the implementation of Industry 4.0 elements depended on the size of the enterprise, but the most important finding was the fact that the implementation of Industry 4.0 elements influenced or had a demonstrable impact on the use of data on the studied processes of the organizations. It was also shown that the separate implementation of selected Industry 4.0 elements will not give organizations the effect they would expect; a comprehensive perception, understanding and implementation of several (or all) elements is required for the resulting effect to be felt. However, it is possible to declare unequivocally and agree with Panetto et al. and Chen et al. that digitalization has a significant impact on manufacturing systems and enhances the ability of an organization to flexibly respond to customer requirements.

Based on the results of the above analysis, it can be clearly stated that the primary challenge for organizations within the Slovak Republic is to learn how to work with the collected data, their appropriate structuring and subsequent archiving (for their use or algorithmizing of solutions in the future). The challenge is to determine the appropriate horizon of usability of the stored data—here we assume differences depending on the industry segment. However, all the challenges combine into a common challenge for 21st-century society: the need to educate and apply data analysts in a broader context.

**Author Contributions:** Conceptualization, M.M. and P.S.; methodology, validation, resources M.M., P.S., S.J. and M.M.-K.; data curation, software M.M. and M.M.-K.; writing—original draft preparation, S.J., P.S., M.M. and M.M.-K.; writing—review and editing, P.S. and M.M.; visualization, M.M. and M.M.-K.; supervision, P.S. All authors have read and agreed to the published version of the manuscript.

**Funding:** The paper is a part of project KEGA No. 018TUKE-4/2022 "Creation of new study materials, including an interactive multimedia university textbook for computer-aided engineering activities". This paper has been published with the support of the Operational Program Integrated Infrastructure within project "Research in the SANET network and possibilities of its further use and development", code ITMS 313011W988, co-financed by the ERDF).

**Institutional Review Board Statement:** Not applicable.

**Informed Consent Statement:** Not applicable.

**Data Availability Statement:** Not applicable.

**Acknowledgments:** The paper is a part of project VEGA No. 1/0721/20 "Identification of priorities for sustainable human resources management with respect to disadvantaged employees in the context of Industry 4.0".

**Conflicts of Interest:** The authors declare no conflict of interest.

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
