# Peer review of "Effective Data Utilization in the Context of Industry 4.0 Technology Integration"

_applsci, doi:10.3390/app122010517_

Round 1

Reviewer 1 Report

The title and abstract of the paper seemed interesting, however the paper covered aspects of Industry 4.0 that have been commonly known - provided facts that have been already published in the past. The Research questions were the only thing that formed some interest. 

The recommendations were hard to follow as there were no comparison provided.

The authors have only addressed the benefits of technologies enabling Industry 4.0, however, there are number of issues that these technologies are susceptible to in terms of data compliance, consistency, etc. which have not been mentioned anywhere. It is suggested to incorporate comparative facts related to these technologies, to provide readers insights of the benefits and limitations of each of the technologies used and what the impact it may have.

The authors should remove repeating the same sentences related to fourth industrial revolution. 

Few other corrections, recommendations have been mentioned below:

L21        Different font size used in abstract

L60-63  Missing Citation

L73-74  Missing Citation

Section 1 (pages 1-4) are continuously repeating the same sentences in terms of fourth industrial revolution / Industry 4.0. Suggest bringing consistency as the repetition may lose readers interest in the paper.

L179 mentions the benefits of using cloud in Industry 4.0. However, there are various limitations in reference to Cloud and Industry 4.0 implementations. Suggest reading the below paper as it aligns with the scope of this research.
Hybrid Cloud SLAs for Industry 4.0: Bridging the Gap. Annals of Emerging Technologies in Computing (AETiC), Print ISSN: 2516-0281, Online ISSN: 2516-029X, pp. 41-60, Vol. 4, No. 5, 2020, Published by International Association of Educators and Researchers (IAER), DOI: 10.33166/AETiC.2020.05.003, Available at SSRN: https://ssrn.com/abstract=3785455

Incorporating the above change will potentially widen the scope of authors research.

Author Response

Dear reviewer,

We would like to thank for your review of the paper and for their valuable advice and attention to detail, which has been incorporated into the paper as follows:

  • L21  Different font size used in abstract – The formal deficiency is removed.
  • L60-63   Missing Citation / L73-74  Missing Citation – Sources cited at the end of paragraphs point to the full paragraph citation from that author/authors.
  • The authors should remove repeating the same sentences related to fourth industrial revolution. Section 1 (pages 1-4) are continuously repeating the same sentences in terms of fourth industrial revolution / Industry 4.0. – The text has been revised, duplications could only be there because of the linking of the different Industry 4.0 technologies in the context of the technology just mentioned.
  • L179 mentions the benefits of using cloud in Industry 4.0. However, there are various limitations in reference to Cloud and Industry 4.0 implementations. Suggest reading the below paper (see link below) as it aligns with the scope of this research. – We agree with the reviewer's opinion (and the content of the recommended publication) regarding cybersecurity in the use of Cloud solutions in industrial entities; however, the focus of this publication was not intended to address the currently known drawbacks and shortcomings of such solutions, but rather to draw attention to the existing and proven benefits of collecting, sorting, processing, and using big data and storing it via Cloud storage.

Yours sincerely,

Author collective

Reviewer 2 Report

The aim of the research was to determine the extent to which individual technologies of Industry 4.0 are applied in the effective utilization of data from specific activities/processes in industrial organizations, to shift the paradigm from traditional organizations to smart organizations. The article is written in clear language and is understandable. The empirical study is simple, but it gives interesting results that can be of interest to the entrepreneurs, as well as the single users of new technologies. However, this article should be slightly corrected. There is a lack a methodical connection between research questions and hypotheses in this paper. There are 5 research questions, and only 2 hypotheses in this study. The hypotheses relate  to the first 2 research questions. Thus, it wonders why other search questions do not have hypotheses (?). In my opinion it should be improved.

It is a good article. Best wishes!

Author Response

Dear reviewer,

We would like to thank you for your review of the paper for their valuable advice and attention to detail, which has been incorporated into the paper as follows:

  • There is a lack a methodical connection between research questions and hypotheses in this paper. There are 5 research questions, and only 2 hypotheses in this study. The hypotheses relate to the first 2 research questions. Thus, it wonders why other search questions do not have hypotheses (?) – RQ3: We did not consider it necessary to test the hypothesis, as the research question was aimed at finding out the current state of implementation of Industry 4.0 technologies. RQ4: We did not consider it necessary to test the hypothesis as the research question was aimed at mapping the use of data collected from individual processes. The question was aimed at finding out the actual state of the matter. RQ5: The research question was formally modified, given that it only mapped the current situation of the use of Industry 4.0 elements and their impact on the use of data from selected processes in organizations.

Yours sincerely,

Author collective

Reviewer 3 Report

The paper may interest readers, but some improvements need to be made.

What is the aim of the paper (research)?

In the Abstract, you state:

This paper aims to identify means of achieving a paradigm shift from traditional organization to smart ones through the use of data in the context of the integration of Industry 4.0 technologies: IoT, Cloud Computing, CPS, and Big Data.

In the chapter Materials and Methods is written:

“The aim of the research was to determine the extent to which individual technologies of Industry 4.0 are applied in the effective utilization of data from specific activities/processes in industrial organizations, to shift the paradigm from traditional organizations to smart organizations.

Does the paper have a different purpose than the research?

How was the literature for the paper selected? Please justify the literature selection and the relationship with the research questions.

Was the questionnaire validated? If so, on what sample and with what result?

I recommend improving the Discussion and Results chapter. I suggest structuring the chapter so that the discussion section has the same structure as the conclusion section, which begins with the sentence in line 505.

Author Response

Dear reviewer,

We would like to thank you for your review of the paper for their valuable advice and attention to detail, which has been incorporated into the paper as follows:

  • Difference between aim of publication in Abstract and in chapter Materials and Methods – Does the paper have a different purpose than the research? – In the abstract of the paper, the objective is stated in general terms, in response to the changing paradigm of the traditional organization towards a smart one through the use of Industry 4.0 technologies. In the Materials and Methods chapter, the objective is refined, tracking the extent of application of selected Industry 4.0 technologies and their impact on the effective use of data. We consider that the above objectives are not divergent, they are focused on the same thing in terms of content - finding out the extent of application of Industry 4.0 technologies and their further impact on the effectiveness of the organization.
  • How was the literature for the paper selected? Please justify the literature selection and the relationship with the research questions. – Literary titles used in the theoretical part represent a summary of knowledge from the latest publications registered in reputable databases or published in journals with a high impact factor. The theoretical knowledge has been summarized to correspond with the content of the research part of the paper. The research questions were formulated for the purposes of the above research (see acknowledgements), and the outputs were statistically evaluated separately for publication purposes.
  • Was the questionnaire validated? If so, on what sample and with what result? – The questionnaire was developed on the basis of a pilot survey (number of respondents 80 - 1 expert per 1 enterprise, members of the industrial council and top managers of the participating industrial enterprises) and on the basis of structured interviews with managers of industrial enterprises (the answer also incorporated in the text of the chapter Materials and Methods).
  • Recommendation to improve the Discussion and Results chapter. – The chapter has been formally and substantively revised, but the core remains unchanged, contains all the important findings of the authors, points to other challenges related to the introduction of Industry 4.0 technologies, which gives adequate space to continue the publication in the near future.

Yours sincerely,

Author collective

Round 2

Reviewer 1 Report

The authors have incorporated only minor changes and included few more citations to support the manuscript.

Author Response

Dear Reviewer,

We would like to thank you for your helpful comments and suggestions aimed at improving the quality of this publication. We have tried to reflect all comments and incorporate the answers into the publication from the Review Report (Round 2):

  • The authors have incorporated only minor changes and included few more citations to support the manuscript. – We believe that all comments leading to the improvement of the quality of the publication have been incorporated into the final version so as not to change the substantive focus of the paper.
  • English language and style are fine/minor spell check required – The article has passed a linguistic review based on recommendation, which was performed by a person with appropriate linguistic qualification.

Yours sincerely,

Author collective

Reviewer 3 Report

Abstract:  

The answer is insufficient. When "the objective is stated in general terms," it must also be mentioned in the paper in "general terms", and then it can be explained in more detail. The abstract can also be published without the core paper.

The method of literature selection must be described in the paper (not only in the answer to reviewer).

Author Response

Dear Reviewer,

We would like to thank you for your helpful comments and suggestions aimed at improving the quality of this publication. We have tried to reflect all comments and incorporate the answers into the publication from the Review Report (Round 2):

  • The answer is insufficient.When "the objective is stated in general terms," it must also be mentioned in the paper in "general terms", and then it can be explained in more detail. The abstract can also be published without the core paper. – Thank you for the comment, the objective in the abstract of the paper has been modified - there is a general focus of the paper as well as a specific objective of the research.
  • The method of literature selection must be described in the paper (not only in the answer to reviewer). – The identified deficiency has been corrected and the requested information has been added to the text of the publication.

Yours sincerely,

Author collective